# Relationship of parenting styles on depression, anxiety, stress and self-esteem of adolescents

Rabina Khadka[1], Anjali Bhatt[1]*, Milan Thapa[2,3,4], Anusha Sharma[5], Manoj Joshi[1], Durga Khadka Mishra[6]

1 Department of Public Health, Manmohan Memorial Institute of Health Sciences, Tribhuvan University, Kathmandu, Nepal, 2 Department of Research and Innovation, Public Health Arena, Kathmandu, Nepal, 3 Nepal Health Frontiers, Kathmandu, Nepal, 4 Space for Adolescence Nepal, Kathmandu, Nepal, 5 School of Public Health, Patan Academy of Health Sciences, Lalitpur, Nepal, 6 Dean, Madan Bhandari Academy of Health Sciences, Hetauda, Nepal

☯ These authors contributed equally to this work.
* bhattanjali80@gmail.com

## Abstract

Globally, fourteen percent of adolescents suffer from mental health issues. Among the many contributing factors to mental health problems, parenting is considered to have a significant effect on both the physical and psychological health of adolescents. This study aimed to assess depression, anxiety, stress, and self-esteem among adolescents and their relationship with parenting style. A community-based cross-sectional study with multistage proportionate sampling was conducted among 583 school-going adolescents from 15th September to 14th October 2022 in Bheemdatt Municipality. This study used the previously validated Depression Anxiety and Stress Scale-21 (DASS-21), Rosenberg Self-Esteem Scale, and Parenting Style and Dimension Questionnaire to assess the status and relationships. Variables were analyzed using a multivariate logistic regression model, and a p-value of <0.05 was considered statistically significant. The prevalence of depression, anxiety, and stress among adolescents was found to be 37.39% (95% CI: 33.46 to 41.32), 42.19% (95% CI: 38.18 to 46.19), and 24.69% (95% CI: 21.18 to 28.19), respectively. Likewise, the mean self-esteem score among adolescents was found to be 22.26±3.56. The authoritative parenting style was found to be significantly associated with DAS and self-esteem among adolescents, while authoritarian parenting style is associated with depression and self-esteem, and permissive parenting style is associated with stress. Social support, and bullying, showed strong effects on the research outcomes. These findings highlight the crucial role of parental involvement and support in shaping adolescent mental health, underscoring the need for positive parenting practices to improve mental health outcomes for adolescents. It is highly recommended that school-based programs focus on raising awareness of mental health issues and offer interventions such as stress management and student counseling. There is a pressing need to effectively monitor and enforce anti-bullying policies within educational settings.

**Data availability statement:** The data underlying the results presented in the study are available from https://doi.org/10.6084/m9.figshare.26535427.v1.

**Funding:** The author(s) received no specific funding for this work.

**Competing interests:** The authors have declared that no competing interests exist.

## Introduction

"Adolescence is the phase of life between childhood and adulthood, from ages 10 to 19. It is a unique stage of human development and an important time for laying the foundations of good health" [1]. During this time, adolescents experience rapid physical, cognitive, psychological, and social changes, including exposure to living conditions, stigma, discrimination, family situation, lack of access to quality health services, which make adolescents vulnerable to mental health problems [2]. These stressors may lead to internalized symptoms such as depression, anxiety, stress, and low self-esteem, which often co-occur and adversely affect psychological functioning and academic performance [2,3]. Globally, 1 in 7 (14%) of adolescents experience mental health disorders, yet most of these conditions remain largely unrecognized and untreated [2]. As per the National Mental Health Survey (NMHS), Nepal 2020, the prevalence of mental disorders among adolescents was recorded at 5.2% [4]. Moreover, within Sudurpashim province, the prevalence of mental disorders was found to be 3.9% [4]. Likewise, a survey by WHO in 2017 reported that among the countries of South East Asia, Nepal had the highest rates of suicidal ideation (14%) and a significant number of suicide attempts (10%) among adolescents, with 5% exhibiting anxiety and 7% experiencing loneliness. This highlights the growing mental health crisis in Nepal, with adolescents disproportionately affected [4–6].

Adolescents' mental health can be broadly explained in terms of depression, stress, anxiety, and self-esteem. These interconnected psychological constructs contribute to the overall well-being of adolescents, shaping their emotional resilience and coping mechanisms. Depression is one of the most prevalent mental health disorders characterized by persistent sadness and loss of interest that co-occurs with anxiety [3]. Likewise, anxiety is the sensation of being nervous or scared over something [3]. Similarly, stress is a typical emotion we experience when we feel pressured, overburdened, or unable to manage [3]. Self-esteem is defined as one's total assessment of one's own thoughts and emotions in connection to oneself as well as their positivity or negativity toward oneself [7]. The interplay among these factors creates a vicious cycle that exacerbates poor mental health outcomes in adolescents.

One of the most important determinants of adolescent mental health is parenting style, which has been found to have a long-lasting effect on the development of teenagers' personalities and other psychological traits, in addition to having a direct impact on their mental health [8–10]. Previous studies have shown that these parenting styles yield varying results on the mental health outcome of a child [11–14]. According to Baumrind 1971, there are three types of parenting styles: authoritative, permissive, and authoritarian [15]. Authoritative parenting shows a particular mix of strong control and supportive promotion of the child's autonomous and independent endeavors; authoritarian parenting is characterized by a controlling nature, dissatisfaction, and distrust towards a child's decision, while permissive parenting is the least demanding, controlling, and generally warm parents [15,16]. Various research studies have shown the influence of parenting style on the emotional development of adolescents [11–13,17,18]. Recent survey conducted in Nepal have shown the association

of the parenting style on the mental well-being of adolescents, with 11.3% of adolescents showing suicidal behaviors and authoritarian parents as strong predictors of such behaviors [12]. In addition, the survey conducted by WHO in 2017 reported parents who were less engaged in their lives had attempted suicide at a rate of 15%, and 6% faced anxiety and loneliness, respectively [6]. These statistics highlight the escalating mental health challenges in Nepali adolescents and underscore the importance of family-level determinants. Given parenting style's significant yet understudied role in adolescent mental health, this factor needs to be explored more, and limited research explaining the correlation between parenting and the above-discussed psychological factors remains a big question to be answered [12]. The available research is conducted in geographical and cultural settings, and findings may not be generalized to adolescents in Sudurpaschim Province [12]. Moreover, few studies have comprehensively examined the relationship between parenting styles and a range of psychological outcomes- namely depression, anxiety, stress, and self-esteem- in a single study. This gap limits our understanding of how parenting approaches shape adolescent mental health in the Nepali context. Likewise, issues and factors of the mental health are barely discussed and identified for Far-western Nepal context, which creates problem of lack of evidence to support programs. Therefore, this study aimed to assess depression, anxiety, stress, and self-esteem in adolescents and their relationship with parenting style in Bheemdatt Municipality, Kanchanpur, Nepal.

## Materials and methods

### Study setting, design and population

The cross-sectional study was conducted among school-going adolescents of Bheemdatt Municipality in Kanchanpur district of Sudurpaschim Province, Nepal. The recruitment period for the study was from 15th September to 14th October 2022. The municipality was named Mahendranagar in honor of the late King Mahendra of Nepal [19]. Later, it was changed to Bhimdatta municipality in honor of the revolutionary farmer leader Bhimdatta Panta, following the country's 2008 republican transition [19]. The municipality has 19 wards with an area of 171.24 km$^2$ (66.12 sq mi) and located 700 kilometers (430 miles) west of Kathmandu and 5 kilometers (3.1 mi) east of the Indian border [19]. Data collection was done in the first week of September 2022. There were 124 schools in the municipality with a population of 5648, of which 10 schools were selected. School-going adolescents were the study population. These school-going adolescents comprised the adolescents of grades 9 and 10. This cross-sectional study was conducted and reported according to Strengthening the Reporting of Observational Studies in Epidemiology (STROBE) guidelines. (Review S1 Checklist. STROBE Statement for more details)

### Sample size calculation and sampling technique

A multistage proportionate sampling technique was used to select the schools and participants randomly. The sample size was estimated using the formula for cross-sectional survey $n = \frac{Z_\alpha^2 pq}{d^2 + \frac{Z_\alpha^2 pq}{N}}$ using prevalence (p) = 0.57 and 1-p (q) = 0.43 [20] and finite population of 5648 adolescents at a confidence interval of 95% and allowable error set at 5%. After the calculation, the minimum sample size was 353. After adjusting the design effect of 1.5, adjusting variance from cluster design, and assuming nonresponse at 10%, the final sample size was 583. Two-stage cluster sampling was used where two separate lists of public and private schools of Bheemdatt Municipality were prepared and then, ten schools (5 public and 5 private schools) were chosen from the list of schools that fall under the criteria. The final sampling distribution was determined through proportionate sampling based on the total number of students in each randomly selected school. The adolescents of grade 9 and 10, who were supported through parental consent, were included in the study. Likewise, schools with less than 50 students were excluded from the study to ensure the desired sample size and feasibility of the study. adolescents without parents were not taken in the study. Later, proportionate sampling was done to calculate the required sample size from each school and the grades and sections included in the study were randomly selected.

## Data collection tools

A self-administered questionnaire, which was used to assess the status of depression, anxiety, stress, and self-esteem in adolescents and its relationship with parenting style, was divided into three sections. The first section includes information on socio-demographic characteristics, family characteristics, and academic characteristics. The second section includes questions to measure perceived parenting style using Parenting Style and Dimension Questionnaire – Short Version (PSDQ) [21]. The Third section includes questions to measure the status of depression, anxiety, and stress among adolescents using the previously validated Nepali version of Depression, Anxiety and Stress Scale – 21 (DASS-21) [22]. The Fourth section includes questions to measure self-esteem levels among adolescents using the Rosenberg Self-esteem Scale [7]. (see S1 Tool for more details)

Parenting Style and Dimension Questionnaire – Short Version (PSDQ) was used to assess the adolescents' perception of parenting style. The Short Version of the PSDQ consists of 32 items rated on a five-point Likert-type scale ranging from 1 (never) to 5 (always) [21]. PSDQ consists of three subscales, namely authoritative, authoritarian, and permissive parenting styles. Each subscale consists of dimensions: authoritative parenting style consists of connection, regulation, and autonomy dimensions; authoritarian parenting style consists of physical coercion, verbal hostility, and non-reasoning/punitive dimensions; permissive parenting style consists of indulgent dimension [21]. The mean score for each subscale is calculated based on the standard scoring approach. To obtain the scoring of each dimension, the mean of each item is calculated [21]. For example, the scoring of the connection dimension is the mean score of the 5 items mentioned under it. Likewise, participants' mean scores for perceived authoritative, authoritarian, and permissive parenting styles were calculated by mean scoring all the items under each dimension [21]. For example, the score for the authoritative parenting style is calculated by the mean score of all 15 items under connection, regulation, and autonomy dimensions. Then, a parent is classified as authoritative if their mean score for the authoritative scale was higher than their mean scores for both authoritarian and permissive scales. This is consistent with PSDQ's standard scoring approach. The Depression, Anxiety, and Stress Scale – 21 Items (DASS-21) is a set of three self-report scales designed to measure the emotional states of depression, anxiety, and stress. Each of the three DASS-21 scales contains 7 items, divided into subscales with similar content. Each item is rated on a 4-point Likert scale, ranging from 0 (indicating "did not apply to me at all") to 3 (indicating "applied to me very much"). Scores for DAS were calculated by summing the scores for the relevant items and multiplying by two [22]. DAS is characterized into normal, mild, moderate, severe, and extremely severe levels based on the recommended cut-offs [22]. (see S1 Table. Study Variables for more details) The Rosenberg Self-esteem Scale, which was used to measure self-esteem in the study, is a 10-item scale that measures global self-worth by measuring both positive and negative feelings about the self. The scale is believed to be uni-dimensional. All items are answered using a 4-point Likert scale format ranging from strongly agree (score: 4) to strongly disagree (score: 1), with the total score ranging from 10 to 40, with reverse scoring of negative items (2, 5, 6, 8, 9) [7]. The scores are later summed for all ten items, and scores >20 are characterized as high self-esteem and scores <20 as low self-esteem. [7,12,23,24] (see S1 Table. Study Variables for more details)

## Data collection procedure and technique

The data collection for the study was started on the 15th September 2022 after the authorization of the ethical approval on the 1st September 2022. Data was collected after the permission from the municipal education division and individual secondary schools. The questionnaire was pretested among 58 school-going adolescents of Kathmandu Metropolitan. Self-administered questionnaires were distributed among the students, and orientation session was provided to students before data collection, and information on mental health issues was provided to students at the end of the session. Written informed consent was taken from all students prior to data collection whereas additional written parental consent was obtained from students below 18 years of age.

## Data analysis

The data entry form was created prior to data collection in IBM SPSS Statistics version 16 and was later pretested, cleaned, and sorted for further analysis. Further, to assess the internal consistency of the tools used, Cronbach's alpha was calculated. The Cronbach's alpha for DASS-21, the Rosenberg self-esteem scale, and PSDQ tool were found to be 0.90, 0.63, and 0.85, respectively. Likewise, data entry errors were minimized through double-entry verification, where two separate entries were compared and discrepancies were resolved by referring to the original questionnaires. The missing data were handled through listwise deletion of entries with more than 20% missing values, and even after deletion, there was sufficient sample size, maintaining a lower impact of missing values.

Descriptive analysis was performed and frequency tables with percentage were generated for categorical variables and mean and standard deviation for continuous variables.

Binary logistic regression was performed to identify associated factors of symptoms of DAS. Firstly, we performed a univariate analysis in which each co-variate (age, sex, ethnicity, type of school, grade, mother's and father's education, mother's and father's occupation, mother's and father's age, income, parent's marital status, type of family, siblings, relationship status, relationship with friends and teachers, involvement in extra-curriculum activity, and bullying) was modeled separately to determine statistical significance. These covariates were based on the theoretical relevance to the outcome of interest and findings from previous literature. We included the confounding variables for analysis, which are commonly recognized as potential influencers of adolescent mental health. Likewise, candidate variables with a p-value less than 0.05 were further analyzed for bivariate and multivariable logistic regression. A p-value of less than 0.05 was deemed statistically significant in multivariable logistic regression, and the strength of association was measured using the adjusted odds ratio (AOR) with a 95% confidence interval. Likewise, point-biserial correlation was used to assess the nature of association between independent (continuous) and dependent variables (dichotomous), and later multivariate logistic regression was done to confirm this association. We choose age and grade as separate entities because of their different developmental nature and impact to the outcome of the study. Multicollinearity for independent variables like age, grade, parents' education, and other potential predictors was assessed through the variance inflation factor (VIF) and tolerance values, with thresholds set at VIF < 5 and tolerance > 0.2, and the results indicated no significant multicollinearity in the model that would compromise regression results. The dataset supporting this analysis is available as supporting information in S1 Data.

## Ethical approval and consent

All the activities associated with the study proceeded after acquiring the required authorization from the Manmohan Memorial Institute of Health Sciences' Institutional Review Committee (MMIHS IRC 875; Ref 79/139; Date: 1 September 2022). Prior approval was sought from the municipality's education sector and the school administration before data collection. Participants were also well-informed about the study. A written informed consent was obtained from the students before the data collection to ensure their willingness to participate and no identification was included in the questionnaire to ensure anonymity and confidentiality. Parental consent was obtained for students who were under the age of 18.

## Results

### Sociodemographic, family, and contextual characteristics of adolescents

The research questionnaires were distributed among 583 school-going adolescents from 5 public and private schools each.

Table 1 demonstrates the demographic and family characteristics of the respondents. The mean age of the respondents was 15.41 ± 1.02, ranging from 10-18 years. Among 583 participants, more than half of the respondents (58.8%) were male in comparison to female respondents (40.8%). With regards to academic characteristics, about forty-two

**Table 1.  Socio-demographic characteristics of respondents.**

| Socio-demographic Characteristics | Category | Frequency | Percentage |
|---|---|---|---|
| Sex | Male | 344 | 59.0 |
| | Female | 239 | 41.0 |
| Age in years | Mean±SD | | 15.41±1.02 |
| | Range | | 10-18 |
| | ≤ 15 | 297 | 50.9 |
| | > 15 | 286 | 49.1 |
| Type of School | Public | 376 | 64.5 |
| | Private | 207 | 35.5 |
| Ethnicity | Brahmin/Chettri | 470 | 82.5 |
| | Janajati | 40 | 6.9 |
| | Dalit | 44 | 7.5 |
| | Thakuri | 22 | 3.8 |
| | Prefer not to say | 7 | 1.2 |
| Grade | Nine | 248 | 42.5 |
| | Ten | 335 | 57.5 |
| Mother's Education | Literate | 493 | 84.6 |
| | Illiterate | 90 | 15.4 |
| Father's Education | Literate | 561 | 96.2 |
| | Illiterate | 22 | 3.8 |
| Family Income (USD) | <107.28 | 150 | 25.7 |
| | 107.28-214.55 | 163 | 28 |
| | 214.55-321.83 | 146 | 25 |
| | >321.83 | 124 | 21.3 |
| Mother's Age | Mean±SD | | 38.42±5.23 |
| | Range | | 25-65 |
| | ≤ 45 | 538 | 92.3 |
| | > 45 | 45 | 7.7 |
| Father's Age | Mean±SD | | 42.75±5.76 |
| | Range | | 31-70 |
| | ≤ 40 | 251 | 43.1 |
| | > 40 | 332 | 56.9 |
| Parent's Marital Status | Married | 573 | 98.3 |
| | Separated | 5 | 0.9 |
| | Divorced | 5 | 0.9 |
| Type of Family | Nuclear | 322 | 55.2 |
| | Joint/Extended | 261 | 44.8 |
| Relationship Status | Single | 503 | 86.3 |
| | Dating | 80 | 13.7 |
| Relationship with Friends | Close | 471 | 80.8 |
| | Not close | 112 | 19.2 |
| Relationship with Teachers | Close | 388 | 66.6 |
| | Not close | 195 | 33.4 |
| Involvement in ECA | Yes | 468 | 80.3 |
| | No | 115 | 19.7 |

*(Continued)*

**Table 1.** (Continued)

| Socio-demographic Characteristics | Category | Frequency | Percentage |
|---|---|---|---|
| If yes, | Sports | 315 | 67.5 |
| | Community Engagement | 11 | 2.4 |
| | Social Works | 53 | 11.3 |
| | Others | 88 | 18.8 |
| Bullying | Yes | 165 | 28.3 |
| | No | 418 | 71.7 |
| If yes, | School | 77 | 46.4 |
| | Online | 32 | 19.3 |
| | Society | 32 | 19.3 |
| | Others | 25 | 15.1 |

ECA: Extracurricular activities.

percent of the respondents belonged to public schools and about fifty-seven percent were in grade 10. Eighty-three percent of the respondents were from Brahmin/Chettri ethnicity. Talking about the parents' educational status, about eighty-four percent of mothers were literate, and about nighty-six percent of the fathers were literate. Likewise, the majority of the students responded that their father as well as mother had attained a secondary level of education. Likewise, 107.28-214.55 USD per month was the majority of the income in most families, that is 28%. The majority of the parents (93%) were married, while the remaining tiny fraction (0.9%) were separated and divorced, respectively. About fifty-five percent of the respondents lived in the nuclear family.

It was noted that the majority of the adolescents (86.3%) were single, and the majority of adolescents (80.8%) described their relationship with friends as close, and about sixty-six percent of adolescents described their relationship with teachers as close. Talking about the involvement in extracurricular activities, it was found that the majority (80.3%) were involved in extracurricular activities (ECA), with 67% involvement in sports activity. Likewise, twenty-eight percent of the adolescents complained of being bullied, of which about forty-six percent of the adolescents were bullied in the school environment.

### Perceived parenting style among adolescents

The results of perceived parenting style and its dimensions are shown in Table 2. It was found that the percentage of authoritative parenting style was 83.2%, authoritarian parenting style was 43.6% and permissive parenting style was 56.6%. The overall mean scores of authoritative, authoritarian, and permissive parenting styles were 4.16±0.64, 2.18±0.83, and 2.83±0.84, respectively.

### Prevalence of depression, anxiety, and stress among adolescents

The prevalence of symptoms of depression, anxiety, and stress was found to be 37.39% (95% CI: 33.46%, 41.32%), 42.19% (95% CI: 38.18%, 46.19%), and 24.69% (95% CI: 21.18%, 28.19%), respectively (Table 3).

### Self-esteem among adolescents

The mean self-esteem score was 22.26±3.56. About the majority of the adolescents (69.3%) showed high self-esteem while the remaining (30.7%) of the adolescents had low self-esteem.

**Table 2. Perceived parenting style among adolescents (n = 583).**

| Parenting style | Dimensions | Mean±SD | Median (IQR) |
|---|---|---|---|
| Authoritative | | 4.16±0.64 | 4.27 (3.80–4.67) |
| | Connection | 4.36±0.67 | 4.60 (4.20–4.80) |
| | Regulation | 4.35±0.70 | 4.60 (4.00–5.00) |
| | Autonomy | 3.77±0.95 | 4.00 (3.20–4.60) |
| Authoritarian | | 2.18±0.83 | 2.00 (1.58–2.58) |
| | Physical Coercion | 2.32±1.13 | 2.00 (1.25–3.00) |
| | Verbal Hostility | 2.45±1.05 | 2.50 (1.50–3.25) |
| | Non-Reasoning/ Punitive | 1.77±0.89 | 1.50 (1.00–2.00) |
| Permissive | Indulgent | 2.83±0.84 | 2.80 (2.20–3.40) |

IQR: Inter-quartile range.

**Table 3. Prevalence of depression, anxiety, and stress among adolescents (n = 583).**

| Level | Depression | | Anxiety | | Stress | |
|---|---|---|---|---|---|---|
| | n | % | n | % | n | % |
| No | 365 | 62.6 | 337 | 57.8 | 439 | 75.3 |
| Mild | 83 | 14.2 | 49 | 8.4 | 49 | 8.4 |
| Moderate | 93 | 16.0 | 101 | 17.3 | 50 | 8.6 |
| severe | 19 | 3.3 | 41 | 7.0 | 34 | 5.8 |
| Extremely severe | 23 | 3.9 | 55 | 9.4 | 11 | 1.9 |

## Association between parenting style and outcome variables

Table 4 represents the significant relationship between parenting style variables and depression, stress, anxiety, and self-esteem. It was found that authoritative parenting style has a low negative correlation with depression (r = −0.21) and anxiety variable (r = −0.20) and negligible negative correlation with stress (r = −0.19) and self-esteem variable (r = −0.16). Likewise, authoritarian parenting style showed a negligible positive correlation with DAS variables ($r_d$ = 0.15, $r_a$ = 0.13, $r_s$ = 0.08), while, low positive relation with self-esteem (r = 0.15). Similarly, the data revealed that permissive parenting style has a negligible positive correlation with DAS variables ($r_d$ = 0.10, $r_a$ = 0.06, $r_s$ = 0.11) and self-esteem (r = 0.07).

## Factors associated with depression

The results of multivariate logistic regression for the factors associated with depression are shown in Table 5. In the final model, the father's age and bullying were found to be significantly associated with depression. Results showed that the odds of depression among adolescents having fathers aged >50 was 2.05 times (AOR = 2.05, 95% CI 1.07–3.93) more

**Table 4. Association between parenting style and outcome variables.**

| Variables | Depression | | Anxiety | | Stress | | Self-esteem | |
|---|---|---|---|---|---|---|---|---|
| | r | p-value | r | p-value | r | p-value | r | p-value |
| **Authoritative** | −0.21 | **<0.0001** | −0.20 | **<0.0001** | −0.19 | **0.0001** | −0.16 | **0.0001** |
| **Authoritarian** | 0.15 | **0.0003** | 0.13 | **0.0019** | 0.08 | 0.0637 | 0.15 | **0.0004** |
| **Permissive** | 0.10 | **0.0132** | 0.06 | 0.1376 | 0.11 | **0.0060** | 0.07 | 0.1042 |

**Table 5. Factors affecting depression, anxiety, stress and self-esteem among adolescents.**

| Characteristics | Depression[1] AOR, 95% CI | Anxiety[2] AOR, 95% CI | Stress[3] AOR, 95% CI | Self-esteem[4] AOR, 95% CI |
|---|---|---|---|---|
| **Sex** | | | | |
| Male | | | Ref | 1.36 (0.94–1.98) |
| Female | | | 2.17 (1.44- 3.27) * | Ref |
| **Age** | | | | |
| ≤ 15 | | | | Ref |
| > 15 | | | | 1.88 (1.18–3.00) * |
| **Grade** | | | | |
| 9 | | Ref | Ref | |
| 10 | | 1.81 (1.27–2.58) * | 1.73 (1.15–2.60) * | |
| **Mother's education** | | | | |
| Illiterate | 1.50 (0.93–2.42) | | | |
| Literate | Ref | | | |
| **Father's education** | | | | |
| Illiterate | | | | 3.79 (0.85–16.86) |
| Literate | | | | Ref |
| **Father's age** | | | | |
| ≤ 45 | Ref | | | |
| > 45 | 2.05 (1.07–3.93) * | | | |
| **Relationship status** | | | | |
| Single | | | | Ref |
| Dating | | | | 1.63 (0.90–2.95) |
| **Relationship with friends** | | | | |
| Not close | 1.17 (0.76–1.84) | 1.99 (1.28–3.10) * | 1.71 (1.07–2.73) * | 2.68 (1.51–4.76) * |
| Close | Ref | Ref | Ref | Ref |
| **Relationship with teacher** | | | | |
| Not close | 1.33 (0.91–1.94) | 1.15 (0.79–1.68) | 1.95 (1.29–2.96) * | 1.08 (0.70–1.65) |
| Close | Ref | Ref | Ref | Ref |
| **Involvement in ECA** | | | | |
| No | | | | 1.67 (1.01–2.76) * |
| Yes | | | | Ref |
| **Bullying** | | | | |
| No | Ref | Ref | Ref | Ref |
| Yes | 2.42 (1.64–3.55) * | 2.26 (1.67–3.62) * | 1.95 (1.27–3.01) * | 2.37 (1.47–3.82) * |

[1]adjusted for education of mother, age of father, relationship with friends, relationship with teacher and bullying. [2]adjusted for grade of study, relationship with friends, relationship with teacher and bullying. [3]adjusted for sex of the respondent, grade of study, relationship status, relationship with teacher and bullying. [4]adjusted for age, grade of study, education of father, relationship with friends, relationship with teacher, involvement in ECA and bullying. OR: Odds Ratio. CI: Confidence Interval. Ref: Reference group. *Statistically significant at p<0.05.*

likely than adolescents with fathers aged ≤ 50. Likewise, the odds of depression among adolescents who reported bullying was 2.42 times (AOR = 2.42, 95% CI 1.64–3.55) more likely than in adolescents who didn't report bullying.

## Factors associated with anxiety

The results of multivariate logistic regression for the factors associated with anxiety are shown in Table 5. The variables that remained in the final model were grade, relationship with friends, and bullying. Grade (AOR = 1.81, 95% CI

1.27–2.58), Relationship with friends (AOR = 1.99, 95% CI 1.28–3.10), Relationship with teachers (COR = 1.49, 95% CI 1.05–2.11) and bullying (AOR = 2.26, 95% CI 1.67–3.62) was found to be significantly associated with symptoms of anxiety.

### Factors associated with stress

Factors associated with stress are presented in Table 5. Multivariate logistic regression between stress and selected independent variables showed that sex (AOR = 0.46, 95% CI 0.31–0.69), grade (AOR = 1.73, 95% CI 1.15–2.60), relationship with friends (AOR = 1.71, 95% CI 1.07–2.73), relationship with teachers (AOR = 1.95, 95% CI 1.29–2.96) and bullying (AOR = 1.95, 95% CI 1.27–3.01) were significantly associated with symptoms of stress.

### Factors associated with self-esteem

The final model of multivariate logistic regression between self-esteem and selected independent variables showed that age (AOR = 1.88, 95% CI 1.18–3.00), relationship with friends (AOR = 2.68, 95% CI 1.51–4.76), involvement in ECA (AOR = 1.67, 95% CI 1.01–2.76) and bullying (AOR = 2.37, 95% CI 1.47–3.82) were significantly associated, as displayed in Table 5.

### Logistic regression for association between parenting styles and outcome variables

Table 6 presents the detailed association between the outcome variables and the predictor variable, parenting style, and socio-demographic variables as control variables. The results showed that authoritative (AOR = 0.40; 95% CI 0.29–0.55) and authoritarian (AOR = 1.45; 95% CI 1.13–1.85) parenting styles were significantly associated with depression; authoritative parenting style (AOR = 0.50; 95% CI 0.37–0.68) was significantly associated with anxiety; authoritative (AOR = 0.54; 95% CI 0.39–0.75) and permissive (AOR = 1.46; 95% CI 1.11–1.91) parenting styles were significantly associated with stress; and authoritative (AOR = 1.52; 95% CI 1.08–2.14) and authoritarian (AOR = 0.73; 95% CI 0.55–0.96) parenting styles were associated with self-esteem. Likewise, females (AOR = 1.85; 95% CI 1.23–2.77), adolescents with a father who is > 45 years old (AOR = 3.84; 95% CI 1.48–9.93), and adolescents who were bullied (AOR = 2.64; 95% CI 1.74–4.01) have higher odds of depression. Similarly, sex (AOR = 1.57; 95% CI 1.06–2.33), grade (AOR = 1.65; 95% CI 1.12–2.43), relationship with friends (AOR = 1.72; 95% CI 1.08–2.73), and bullying (AOR = 2.70; 95% CI 1.78–4.08) were significantly associated with anxiety. Grade (AOR = 1.69; 95% CI 1.07–2.66), relationship with teachers (AOR = 1.74; 95% CI 1.12–2.72), and bullying (AOR = 1.99; 95% CI 1.27–3.15) were predictors of stress among adolescents. While, age (AOR = 1.77; 95% CI 1.07–2.93), relationship with friends (AOR = 0.39; 95% CI 0.22–0.71), and bullying (AOR = 0.42; 95% CI 0.26–0.69) were significantly found to be associated with low self-esteem.

## Discussion

The present study elucidates the status of the psychological outcomes, including depression, anxiety, stress, and self-esteem, among adolescents of Bheemdatt municipality, Kanchanpur, Nepal, and their intricate associations with parenting styles. Notably, depression, anxiety, and stress were highly prevalent mental health burdens among adolescents. Our findings underscore the nuanced influence of parenting approaches, with the authoritative parenting style exhibiting a low negative correlation with DAS and self-esteem while the authoritarian and permissive parenting style showed a negligible positive correlation across these psychological dimensions. Results from logistic regression revealed that authoritative parenting style is protective against DAS, but associated with low self-esteem; authoritarian parenting style increases the odds of depression and increases self-esteem; and permissive parenting style is associated with higher stress. Likewise, external factors like gender, bullying, and social relationships further exacerbated the risks of psychological distress.

Study results revealed the prevalence of depression, anxiety, and stress to be 37.39%, 42.19%, and 24.69%, respectively. Past studies conducted by Sharma et al. (depression: 41.6%, anxiety: 56.9%) and Karki et al. (depression: 56.5%,

**Table 6. Logistic regression results for association between parenting styles and outcome variables.**

| Characteristics | Depression AOR, 95% CI | Anxiety AOR, 95% CI | Stress AOR, 95% CI | Self-esteem AOR, 95% CI |
|---|---|---|---|---|
| Authoritative | 0.40 (0.29–0.55) * | 0.50 (0.37–0.68) * | 0.54 (0.39–0.75) * | 1.52 (1.08–2.14) * |
| Authoritarian | 1.45 (1.13–1.85) * | 1.21 (0.96–1.55) | 1.19 (0.90–1.54) | 0.73 (0.55–0.96) * |
| Permissive | 1.15 (0.90–1.46) | 1.07 (0.85–1.35) | 1.46 (1.11–1.91) * | 0.79 (0.61–1.03) |
| **Sex** | | | | |
| Male | Ref | Ref | Ref | Ref |
| Female | 1.85 (1.23–2.77) * | 1.57 (1.06–2.33) * | 2.30 (1.46–3.62) | 1.25 (0.83–1.87) |
| **Age** | | | | |
| ≤ 15 | Ref | Ref | Ref | Ref |
| > 15 | 1.26 (0.76–2.10) | 0.97 (0.58–1.61) | 1.25 (0.70–2.24) | 1.77 (1.07–2.93) * |
| **Grade** | | | | |
| 9 | Ref | Ref | Ref | Ref |
| 10 | 1.10 (0.74–1.64) | 1.65 (1.12–2.43) * | 1.69 (1.07–2.66) * | 1.00 (0.66–1.51) |
| **Type of School** | | | | |
| Public | 1.10 (0.73–1.67) | 1.33 (0.89–1.98) | 1.12 (0.70–1.78) | 1.52 (0.99–2.33) |
| Private | Ref | Ref | Ref | Ref |
| **Mother's education** | | | | |
| Illiterate | 1.37 (0.77–2.43) | 1.07 (0.61–1.90) | 0.99 (0.52–1.86) | 1.00 (0.53–1.90) |
| Literate | Ref | Ref | Ref | Ref |
| **Father's education** | | | | |
| Illiterate | 1.02 (0.38–2.80) | 0.72 (0.27–1.97) | 0.37 (0.09–1.48) | 0.28 (0.06–1.39) |
| Literate | Ref | Ref | Ref | Ref |
| **Father's age** | | | | |
| ≤ 45 | Ref | Ref | 1.94 (0.62–6.02) | Ref |
| > 45 | 3.84 (1.48–9.93) * | 2.19 (0.88–5.42) | Ref | 1.06 (0.38–2.90) |
| **Mother's age** | | | | |
| ≤ 45 | 2.07 (0.79–5.40) | 1.82 (0.73–4.49) | Ref | 1.70 (0.64–4.51) |
| > 45 | Ref | Ref | 1.68 (0.61–4.64) | Ref |
| **Parents' marital status** | | | | |
| Separated/Divorced | 0.35 (0.06–2.05) | Ref | 2.85 (0.71–11.46) | Ref |
| Married | Ref | 1.22 (0.29–5.07) | Ref | 1.27 (0.29–5.45) |
| **Family income** | | | | |
| <=30 | Ref | Ref | 1.21 (0.78–1.87) | 1.02 (0.53–1.90) |
| >30 | 1.14 (0.77–1.69) | 0.93 (0.63–1.36) | Ref | Ref |
| **Relationship status** | | | | |
| Single | 1.05 (0.60–1.81) | 1.51 (0.88–2.60) | 1.09 (0.58–2.02) | 1.58 (0.84–2.94) |
| Dating | Ref | Ref | Ref | Ref |
| **Relationship with friends** | | | | |
| Not close | 0.94 (0.58–1.51) | 1.72 (1.08–2.73) * | 1.51 (0.91–2.49) | 0.39 (0.22–0.71) * |
| Close | Ref | Ref | Ref | Ref |
| **Relationship with teacher** | | | | |
| Not close | 1.11 (0.74–1.68) | 1.03 (0.69–1.53) | 1.74 (1.12–2.72) * | 0.98 (0.63–1.54) |
| Close | Ref | Ref | Ref | Ref |
| **Involvement in ECA** | | | | |
| No | 1.15 (0.72–1.83) | 1.17 (0.75–1.84) | 1.42 (0.87–2.32) | 0.63 (0.38–1.06) |
| Yes | Ref | Ref | Ref | Ref |

*(Continued)*

**Table 6.** (Continued)

| Characteristics | Depression AOR, 95% CI | Anxiety AOR, 95% CI | Stress AOR, 95% CI | Self-esteem AOR, 95% CI |
|---|---|---|---|---|
| **Bullying** | | | | |
| No | Ref | Ref | Ref | Ref |
| Yes | 2.64 (1.74–4.01) * | 2.70 (1.78–4.08) * | 1.99 (1.27–3.15) * | 0.42 (0.26–0.69) * |

OR: Odds Ratio; CI: Confidence Interval; Ref: Reference group; *Statistically significant at p < 0.05.

anxiety: 55.6%) reported higher rates [20,25]. While Gautam et al.'s study in rural Nepal found lower depression rates (27%) [26]. Anxiety findings were particularly variable, ranging from 46.5% to 55.6%, higher than the existing study [20,27]. Stress symptoms showed more consistency across studies, 27.5% to 32.9% [20,25]. The prevalence of depression was found to be higher compared to the study conducted in the rural setting of Nepal. One of the possible explanations for this could be that the data was collected during the very exam season. During this period, students may have struggled academically and displayed symptoms of depression, anxiety, and stress. Likewise, the majority of the students in the study were grade 10 students, who were under constant pressure with the approaching SEE exams. Among South Asian countries, the prevalence of depression reported by our study is lower than the studies conducted in India, Bangladesh, and China [28–31]. On the contrary, the prevalence of anxiety as reported by the current study is in line with the studies conducted in China and Vietnam, but higher than that of Sri Lanka and lower than that of India [28,30,32,33]. Likewise, the prevalence of stress is comparable with the study conducted in Manipur, India but lower than that of a similar study conducted in Chandigarh, India [29,34]. These findings suggested the increasing burden of DAS among school-going adolescents in South Asia. The mean self-esteem score in this study was found to be 22.26 ± 3.56. This was in line with the study conducted in Nigeria (24.0 ± 3.3) [35]. However, this differs from the research findings of Banstola et al. (16.59 ± 0.16) [12] and the study in Iran (38.49 ± 6.55) [36]. The possible reason for this might be the difference in study setting and population.

Our results highlight the subtle yet distinct effects of parenting styles; adolescents who perceived their parents as authoritative have lower odds of depression (AOR = 0.40; 95% CI 0.29–0.55), anxiety (AOR = 0.50; 95% CI 0.37–0.68), and stress (AOR = 0.54; 95% CI 0.39–0.75). These results are consistent with previous studies [37,38]. This outcome is corroborated by that of Prativa et al., which indicated a negative correlation between authoritative parenting style and depression in adolescents, that is, more authoritative parenting style is associated with less depression in adolescents [17]. Similar results were found in the study conducted in Japan in later adults [39] and in Indonesia, in high school students [40]. However, some prior research by Mishra et al. reported higher odds of social anxiety among adolescents with authoritative parents [41]. On the other hand, authoritarian parenting style was associated with higher odds of depression (AOR = 1.45; 95% CI 1.13–1.85). This finding was corroborated by the existing study that revealed the effect of negative parenting on adolescent mental health [37,38,42]. In similar manner, permissive parenting style was significantly associated with higher odds of stress (AOR = 1.46; 95% CI 1.11–1.91). Results from Sanjeevan et al. indicated that authoritative parenting style correlated with a lower level of DAS, while neglectful parenting style correlated with a higher level of DAS [14]. Likewise, A cross-sectional study on US adolescents revealed parental care and control were associated with mental health disorders among adolescents, with involved parenting having a positive role and neglectful parenting having a negative role [43]. Similar study in India among school-going adolescents showed authoritative parenting style positively impact mental health compared to authoritarian and permissive parenting styles [44]. Existing research also showed the influential role of parenting styles on adolescent psychological well-being [45]. Research findings showed authoritative parenting style (AOR = 1.52; 95% CI 1.08–2.14) was significantly associated with low self-esteem, while authoritarian parenting style (AOR = 0.73; 95% CI 0.55–0.96) showed positive effects on self-esteem. This differs from the existing study conducted on the urban high school students of Nepal by Banstola et al., which reported a positive association between

parental authoritativeness and self-esteem [12]. This discrepancy might be due to differences in the study setting and population. Likewise, a study in Indonesia showed a beneficial effect of authoritative parents on self-esteem and a negative effect of authoritarian parents on self-esteem, which differs from the research findings [40]. Research study by Milevsky et al. revealed the supportive role of an authoritative parenting style in increasing self-esteem [11]. The possible explanation for these results may be due to the nature of parenting style. It has been evident that an authoritative parenting style is characterized by parents who tend to display love and warmth, and high responsiveness towards the children. This reason might have resulted in a decrease in the symptoms of DAS among school-going adolescents [43]. However, the authoritative parenting style showed a negative correlation with self-esteem which resulted in a decrease in self-esteem. The result might be explained by the fact that the control nature of authoritative parents might increase the high expectations towards children as a result this will decrease self-esteem in adolescents. Likewise, authoritarian and permissive parenting styles showed negative association with DAS and self-esteem. This relationship may partly be explained by the fact that authoritarian parents try to keep their children in the controlled environment pressurizing their children which will increase DAS [43,46]. Likewise, permissive parents are less involved with their children which might have increased DAS [46].

In the current study, it was found that female adolescents were more likely to suffer from symptoms of stress than male adolesents. The findings of the study are consistent with the research of Sharma et al. which showed higher symptoms of stress in females [20]. This is inconsistent with the findings of the previous research studies [32,47–50]. One of the possible explanations could be a different approach to parenting for males and females, leading to variation in mental health status, and females are often exposed to family burden and social stigma [51,52]. Similarly, the current study showed higher self-esteem scores in male adolescents than in female adolescents. These results agree with the findings of a study by Banstola et al. [12]. Likewise, study in Iran showed higher self-esteem in male adolescents than female adolescents [36]. On the contrary, another study in Iran reported higher mean scores of girls than boys [18].

Mothers' and fathers' education wasn't found to be significant with the DAS and self-esteem among adolescents. However, existing research studies by Gautam et al. [28] and Bhattrai et al. [53] showed the significant association of depression with parental education. Likewise, the research findings of Alami et al.. [36] reflected significant association between the father's education and the self-esteem of adolescents. Relationships with friends showed a significant association with anxiety, stress, and self-esteem, and relationships with teachers showed a significant association with stress. It was found that the odds of depression, anxiety, stress, and low self-esteem among adolescents having no close relationship with friends were 1.17 times, 1.99 times, 1.71 times, and 2.68 times more likely than adolescents with close relationships with friends, respectively; however, depression wasn't significant. However, some research conducted in Nepal and China highlighted the significant impact of a poor relationship with friends on depression among adolescents [26,53,54]. Likewise, research findings are consistent with the existing research, which showed an association of anxiety with a lack of peer relations [55–57]. Studies conducted in Nepal reported a lack of support from friends, family, and teachers as a predictor of anxiety [58,59]. Similarly, another study showed a positive correlation between self-esteem and social support, consistent with the study findings [60]. Consistent with research findings, Keane et al. showed teacher-student closeness predicted less aggressive and stressful behaviors in adolescents [61]. A possible explanation might be the fact that a lack of social support results in a lack of confidence to tackle issues and worry about small problems, which eventually leads to early signs of DAS and low self-esteem.

Bullying was also found to be significant among adolescents and was associated with increased levels of DAS and a decrease in self-esteem among adolescents. Likewise, the odds of depression, anxiety, stress, and low self-esteem among adolescents who reported bullying were 2.42 times, 2.17 times, 1.95 times, and 2.37 times more likely than in adolescents who didn't report bullying, respectively. It is consistent with the research findings of Karki et al., which showed the symptoms of depression were 2.84 times more likely among the students who were bullied electronically, symptoms of anxiety and stress were 1.36 and 1.48 times more likely in adolescents who were bullied on school property, respectively [20]. Likewise, a similar study in Chinese high school students revealed that bullying victimization results in an increased

level of social anxiety with the mediating role of self-esteem [62]. Similarly, another study showed consistent findings which confirmed that bullying victims have higher depressive symptoms, lower self-esteem, and higher suicide tendencies than non-victims [63]. Existing research on students also confirmed the significant association between stress and bullying [64]. The students who were bullied repeatedly experience abuse and threats, which exacerbates the feeling of fear and loneliness. This causes emotional distress in adolescents, which leads to increased DAS symptoms and low self-esteem.

The findings of the study should be interpreted in light of its limitations. This study took the perspective of adolescents on the perceived parenting style. This perception might differ among the parents. Future studies should try to include the perspectives of both parents and children. Likewise, maternal and paternal parenting style was found to be different and can interfere with the findings, so future studies should consider taking the perception of parenting style from each parent. The measures used in the study relied on the adolescents' self-reports on their mental status, which may introduce recall bias on this sensitive topic. Parental perspective on this topic may include some dynamic interactions; future studies should include multi-informants to triangulate the data. Although validated in Nepali, PSDQ was developed in the western context. Cultural variations in the parenting norms might affect instrument validity; this should be considered for future study. Likewise, the sample included school-going adolescents in one municipality, limiting generalizability to out-of-school adolescents or adolescents of other regions of Nepal. Furthermore, the exclusion of schools with very few enrollments may limit the generalizability of our findings to adolescents within such educational settings. Future research should try a robust methodological approach to further explore the potential predictors of the study.

## Conclusion

This study planned to assess depression, anxiety, stress, and self-esteem in adolescents and its relationship with parenting style in Bheemdatt Municipality, Kanchanpur. About one-third of adolescents showed symptoms of depression, almost half of the adolescents showed symptoms of anxiety, and about one-fourth of adolescents showed symptoms of stress in Bheemdatt Municipality. Likewise, the majority of the adolescents showed high self-esteem, and about one-third of adolescents showed low self-esteem. Importantly, authoritative parenting style was found to be protective against DAS but predicted lower self-esteem among adolescents, while authoritarian parenting style is associated with a higher rate of depression and self-esteem, and permissive parenting style is associated with higher stress. This shows the important role of parenting behavior in the prediction of the mental health status of adolescents, with positive parenting behavior showing a positive impact on the behavior of the child. Importantly, the social support system and bullying emerged as strong predictor of adolescent well-being, with poor social relationships and bullying victimization substantially increasing DAS and decreasing self-esteem. Thus, improving adolescents' perceived level of social support may alleviate the psychological distress among adolescents. Therefore, prevention and control activities such as school-based counseling focusing on bullying will help to reduce psychological distress and improve self-esteem among adolescents. Our findings also advocate for integrating mental health screening in the school health programs, particularly in underserved regions like Sudurpaschim Province. This study underscores adolescent mental health is shaped by parenting practices and social environment. Nepal's adolescent suicide rates are among the highest in Southeast Asia. By identifying modifiable family-level factors like parenting style alongside systemic issues like bullying, our study provides strategic points to mitigate this problem. This serves as the evidence and foundation data in the context of Nepal and important document for policymakers to review parenting guidelines to Nepali cultural norms. Hence, it is concluded that parenting styles are significantly associated with adolescents' psychological distress and self-esteem, with the supporting role of study covariates and social support systems.

## Supporting information

**S1 Table. Study variables.** https://doi.org/10.6084/m9.figshare.30490772
(PDF)

**S1 Tool. Study tool.** English Questionnaire. https://doi.org/10.6084/m9.figshare.30490790 (PDF)

**S1 Data. Study dataset.** Final version of study dataset. https://doi.org/10.6084/m9.figshare.30490658 (XLSX)

**S1 Checklist. STROBE statement.** This is the Strengthening the Reporting of Observational Studies in Epidemiology (STROBE) Statement guidelines for reporting observational studies. https://doi.org/10.6084/m9.figshare.30490805 (DOCX)

## Acknowledgments

Foremost, we express our sincere gratitude towards the students for their cooperation and support during the study. Special thanks to Bheemdatt municipality for granting permission to conduct the study. Grateful to the school administration and teachers for their assistance during data collection. Above all sincere gratitude towards and Department of Public Health, Manmohan Memorial Institute of Health Sciences for their unwavering leadership and assistance.

## Author contributions

**Conceptualization:** Rabina Khadka, Anjali Bhatt.

**Data curation:** Anjali Bhatt.

**Formal analysis:** Anjali Bhatt, Milan Thapa.

**Investigation:** Anjali Bhatt.

**Methodology:** Rabina Khadka, Anjali Bhatt.

**Project administration:** Anjali Bhatt.

**Supervision:** Rabina Khadka, Durga Khadka Mishra.

**Visualization:** Anjali Bhatt.

**Writing – original draft:** Rabina Khadka, Anjali Bhatt.

**Writing – review & editing:** Rabina Khadka, Anjali Bhatt, Milan Thapa, Anusha Sharma, Manoj Joshi, Durga Khadka Mishra.

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
