## [Decision Letter · Decision Letter 0]

14 Feb 2025

Dear Dr. Bhatt,

Thank you for submitting your manuscript to PLOS ONE. After careful consideration, we feel that it has merit but does not fully meet PLOS ONE’s publication criteria as it currently stands. Therefore, we invite you to submit a revised version of the manuscript that addresses the points raised during the review process.

Dear Author

Thank you for your intention to publish an article in Plos One journal.

The opinions of two reviewers have arrived.

I would suggest to read the comments of the reviewers carefully, and refer to them.

The two reviewers have comments regarding various parts of the article, after completing corrections and addressing the comments, we will be able to consider the article after you will resubmit it.

We look forward to receiving your revised manuscript.

Kind regards,

Gal Harpaz, Ph.D.

Academic Editor

PLOS ONE

Journal Requirements:

Additional Editor Comments:

Dear Author

Thank you for your intention to publish an article in Plos One journal.

The opinions of two reviewers have arrived.

I would suggest to read the comments of the reviewers carefully, and refer to them.

The two reviewers have comments regarding various parts of the article, after completing corrections and addressing the comments, we will be able to consider the article after you will resubmit it.

Best regards

Reviewers' comments:

Reviewer's Responses to Questions

**Comments to the Author**

1. Is the manuscript technically sound, and do the data support the conclusions?

Reviewer #1: Yes

Reviewer #2: Partly

2. Has the statistical analysis been performed appropriately and rigorously?

Reviewer #1: Yes

Reviewer #2: No

3. Have the authors made all data underlying the findings in their manuscript fully available?

Reviewer #1: Yes

Reviewer #2: Yes

4. Is the manuscript presented in an intelligible fashion and written in standard English?

Reviewer #1: Yes

Reviewer #2: Yes

Reviewer #1: Dear Editor-in-Chief,

Plos One,

It is an honor for me to have been considered for the review of this research study. The topic is both interesting and novel, and the authors have undertaken a rigorous process in constructing the theoretical framework, methodology, results, discussion, and conclusions. However, I believe that some minor revisions could enhance the overall quality of the study. Below, I provide my suggestions:

1. Title: The title of the article is: “Influence of parenting styles on depression, anxiety, stress, and self-esteem of adolescents.”

Comment: I suggest changing the word “influence,” as cross-sectional studies cannot claim "influence"; this is typically associated with experimental or longitudinal studies. Using this term may confuse readers.

2. Abstract: Its ok.

3. Introduction: The introduction needs to maintain a logical argument and structure that is easy to follow for the reader. For example, in line 47, risk factors are introduced, only to be mentioned again in line 57. The same happens with mental health issues: they are introduced in line 49 and end in line 53, only to reappear in line 58, and again, parenting styles are introduced in line 55. Further, mental health issues are mentioned again between lines 58 and 65, and parenting styles are reintroduced in line 65.

Comment: The introduction could be structured more effectively to avoid repetition and improve readability. It should begin by clearly defining the population and the issue (e.g., "Adolescence is…"), then introduce the dependent variables (DAS, self-esteem), and finally, explain the independent variable as an explanatory framework for the dependent variables. In line 76, the limited research on this topic is mentioned, but it would be helpful to add a brief paragraph discussing the limitations of prior research and how this study addresses that gap through its objectives.

4. Method:

Comment: In line 106, what are the inclusion criteria?

In the section on Data collection tools (line 127), it is unclear how self-esteem is coded. Are there reverse-scored items? Is it summed or subtracted?

Was unidimensionality checked?

In Data analysis, it would be important to detail how missing data, outliers, and errors were handled. Was the internal consistency of the instruments verified?

It would also be helpful to specify how covariates were selected. Was any preliminary testing (such as correlation analysis or analysis of potential confounding variables) done before selecting the variables for univariate analysis?

In the multivariable logistic regression analysis, was multicollinearity assessed?

5. Results:

Comment: It is recommended to report the results directly in percentage terms rather than saying “two-thirds” or “five-sixths.”

In line 177, the range 15,000-30,000 is mentioned. It would be helpful to include the equivalent in USD, as readers are not focused solely on Nepal.

In line 198, it is stated that the scale consists of three subscales with their dimensions; it would be useful to reference this scale earlier, such as in line 119.

In Table 3, a categorization of depression, anxiety, and stress levels is mentioned, but it is not addressed in the data analysis. Which authors support this categorization, and are there any established cutoffs in Nepal?

Similarly, with self-esteem, the cutoff points for high and low self-esteem are not mentioned. This should be addressed in the data analysis.

In Table 4 (line 229), global scores are calculated. It would be beneficial to elaborate on this when referencing the instruments used.

6. Discussion:

In general, the discussion is well-developed, but it needs better structure, as it currently lacks a clear logical flow. As a reader, it is easy to get lost in the content. I would recommend organizing it around the study's objectives. The objectives are: "This study aimed to assess depression, anxiety, stress, and self-esteem in adolescents and their relationship with parenting style in Bheemdatt Municipality, Kanchanpur, Nepal."

7. Conclusions:

In this sentence: "Hence, it is concluded that parents play an influential role in promoting the mental health of their adolescent children with the supporting role of study covariates and social support systems."

Comment: Care should be taken with the language used, as "influence" cannot be concluded here; rather, "association" or "relationship" should be used. Also, referring to "mental health" is ambiguous, as it is often understood in a bipolar model (presence and absence). This study actually addresses psychological distress and self-esteem, which are well-supported topics in the literature.

8. References:

Comment: In this section, references should be adjusted to meet Vancouver style guidelines (e.g., World Health Organization. Adolescent health. Available: https://www.who.int/health-topics/adolescent-health#tab=tab_1) for various types of sources: books, book chapters, original articles, and reports. To facilitate this, I recommend using a reference manager such as Mendeley and downloading the Vancouver style from Plos One.

9. Supplementary Files:

It is necessary to mention the supplementary files in the text. For example, when discussing the instruments, it should be noted that more details can be found in Table A of the supplementary files. Additionally, it is important to cite and reference the authors used for the cutoff scores.

Overall, the authors have done an excellent job, and the comments provided aim to contribute to improving the quality of the study. Congratulations on your work!

Reviewer #2: Thank you for inviting me to review the manuscript “Influence of parenting styles on depression, anxiety, stress and self-esteem of adolescents". The study investigates the relationship between parenting styles and depression, anxiety, stress, and self-esteem among adolescents in Nepal. While the topic is certainly important for adolescents' mental health, I think that a major work needs to be done for it to be suitable for publication.

INTRODUCTION:

1) The justification of the study is not very strong, especially when there are literature review and studies already conducted on the topic. How will this study make a difference and contributions?

METHODS:

1) What was the reasoning behind excluding schools with less than 50 children? What were the criteria of selecting the 10 schools out of the 124 schools?

2) What is the meaning of high scores in DASS, PSDQ, and Rosenberg Self-esteem Scale? What are the cut-off scores to define different levels of MH in Table 3 , 4, and 5?

3) Were all students attending the 10 schools participated in the survey? If not, what was the participation number/rate?

4) Data analysis: what were the covariates you tested and used in the multivariate logistic?

5) Why did you not use multilevel model when your data are clearly clustered? Did you measure the ICC?

RESULTS:

1) “the percentage of authoritative parenting style was 83.2%) What were the criteria to define authoritative parenting style? The same with the other two style. In addition, what is the connection between parenting style and dimensions in Table 2? More explanations are needed for readers who are not familiar with the topic.

2) Association between parenting style and outcome variables: analysis to measure this was not mentioned in the statistical analysis section. what sort of analysis did you use? For the variables, did you use the scores or categories?

3) Table 5: did you use the parenting style one of the variables in the model? Your research question was: investigating the relationship between parenting style and MH. Table 5 shows the association between demographic factors and MH, which was not part of the research question. Instead, Table 5 should report results from logistic regressions using parenting style as the main predictor and the demographic factors as the covariates (control variables).

4) Table 5: have you tried to run a logistic regression predicting depression (or anxiety/stress/self-esteem) using the parenting style as the main predictor and you used all of the characteristics listed in Table 5 (instead of choosing only those that were significant in bivariate analysis?

5) Have you check the multicollinearity issue (e.g., age and grade would be highly correlated)? Why did you choose age and grade separately, instead of using one of them as predictor?

DISCUSSION:

1) There are close to 2 pages of discussion about the prevalence of depression, anxiety, stress, and self-esteem (including comparisons with different countries) – page 18 to 20. Although this is useful, I feel that the discussion could be made more concise., which will make the paragraph easier to digest and the readers can get the message easily.

2) Starting from line 346 is the discussion about demographic factors influence on MH, which was not mentioned in the research question – so, why was there a bigger proportion discussing the sociodemographic factors compared to the discussion about the parenting style?

3) Line 346-355, discussion about mothers' and fathers' education: they were not significant predictors of MH in Table 5 - so I would expect the discussion would be the reasons for not finding the effect, rather than sayng that the risk of depression was higher in adolescents with mothers without any formal education (when the OR was not significant)

4) Line 356 – 359: only ORs for Anxiety, stress, and self-esteem that were significant – but the mention gave the impression that OR for depression was also significant. Also, the OR for depression was not 1.52 but 1.17.

5) Line 356-373: I would like to see the relationship with teachers to be discussed as well

6) Limitations: I would like to see more limitations being identified, especially in relationship to the study design and methods.

7) What are the implications and contributions of the study? Why do we need to care about this study?

**Do you want your identity to be public for this peer review?** For information about this choice, including consent withdrawal, please see our Privacy Policy

Reviewer #1: No

Reviewer #2: No

---

## [Author Response · Author response to Decision Letter 1]

31 Jul 2025

Dear Editors,

Thank you for the helpful and constructive comments from reviewers and editorial team. We have carefully revised the manuscript and addressed all the comments. A detailed point-by-point response has been uploaded as a separate file named "Response to Reviewers".

We hope revised version meet your expectations. Thank you.

Best regards,

Anjali Bhatt

(On behalf of all authors)

---

## [Decision Letter · Decision Letter 1]

9 Oct 2025

Dear Dr. Bhatt,

We look forward to receiving your revised manuscript.

Kind regards,

Jenna Scaramanga

Staff Editor

PLOS ONE

Journal Requirements:

Additional Editor Comments:

Your manuscript has been evaluated by two reviewers, and their comments are available below. Reviewer 3 requests some clarifications and refinements about the data analysis. Please carefully revise your manuscript to address the points raised.

"The possible explanation must be the difference in parenting between male and female adolescents which can result in differences in the mental status of the adolescents. Likewise, females are more likely to be exposed to the family burden and social stigma."

The last line of this is missing a reference to support it. Additionally, we would suggest rephrasing the opening to "One possible explanation could be" as there are other possible explanations for these results, which isn't clear in the current wording.

Reviewers' comments:

Reviewer's Responses to Questions

**Comments to the Author**

Reviewer #1: All comments have been addressed

Reviewer #3: All comments have been addressed

2. Is the manuscript technically sound, and do the data support the conclusions?

Reviewer #1: Yes

Reviewer #3: Yes

3. Has the statistical analysis been performed appropriately and rigorously?

Reviewer #1: Yes

Reviewer #3: Yes

4. Have the authors made all data underlying the findings in their manuscript fully available?

Reviewer #1: Yes

Reviewer #3: Yes

5. Is the manuscript presented in an intelligible fashion and written in standard English?

Reviewer #1: Yes

Reviewer #3: Yes

Reviewer #1: Dear Editor in Chief

I sincerely appreciate the opportunity to review the manuscript titled “Relationship of parenting styles on depression, anxiety, stress, and self-esteem of adolescents”, conducted in a socially and culturally diverse context such as Nepal.

Overall, I have observed that, in the introduction, method, data analysis, results, discussion, and conclusions, the authors have substantially improved the manuscript by carefully addressing each of the earlier comments. These revisions have resulted in greater consistency, coherence, and clarity in the theoretical implications of the study.

This study presents important and relevant research. Compared to the first version, the current manuscript shows significant progress, with the necessary improvements successfully implemented.

Thank you for your thorough work, and congratulations to the authors for their dedication and the quality of this research.

Reviewer #3: This is an interesting study assessing the relationship between mental health in adolescents and parenting styles.

Some minor comments to add clarity to the reporting.

1. Sample size, can the authors indicate, whether on average it was expected to recruit approx 58 participants from either public or private - would you expect a similar distribution or more in one type than another. Only because there is statement regarding exclusion of school that had <50 student, would this contribute to some selection bias?

2. For ease of reading, suggest to include in the text, what p and q is? How was the design effect obatained and what was the variance cluster

3. The essence of univariate analyses, if the main goal is to understanfding which confounding variables go into the multivariate model, should be to assess significance. NOt sure assessing odds only in univariate is appropriate. Line 225, 226 - can this include line 229.

4. Was potential mediators also looked into? i.e impact of income as a confounder on the parenting styles and adolescent mental health, or even together with influecing the type school you went to, i.e private or public?; considered as some exploratory analyses.

5. The income reported in dollars is, that monthly or annual? it would be good to clarify? Does it 72% did not have an income? or the range is outside this. It would be good to give context, is this low, medium or high earnings?

6. Table 1: Report Range of Age ( students, mothers, fathers) in the table?

7. Table 1: Education level, does illiterate mean (no education?)

8. Table 1: include for words for ECA in table or footnote so reader is not searching for what ECA means.

9. Table 2: Helps to present ranges, assuming these are normally distributed, suggested to also present median (IQR) too.

**Do you want your identity to be public for this peer review?** For information about this choice, including consent withdrawal, please see our Privacy Policy

Reviewer #1: No

Reviewer #3: No

---

## [Author Response · Author response to Decision Letter 2]

16 Oct 2025

Manuscript ID: PONE-D-24-44012R1

Title: Relationship of parenting styles on depression, anxiety, stress and self-esteem of adolescents

We sincerely thank the editor and reviewers for their time and valuable feedback. We are grateful for the overall positive reception of our revised manuscript. We have carefully addressed the points raised in the letter, and changes have been made to improve the quality of our work. You can find a point-to-point detailed response below. Likewise, all the changes have been incorporated into the manuscript, and revisions are highlighted in the file named "Revised Manuscript with Track Changes".

Editor

Comment: Additionally, the journal office requests that you revise lines 417 - 420: "The possible explanation must be the difference in parenting between male and female adolescents which can result in differences in the mental status of the adolescents. Likewise, females are more likely to be exposed to the family burden and social stigma."

The last line of this is missing a reference to support it. Additionally, we would suggest rephrasing the opening to "One possible explanation could be" as there are other possible explanations for these results, which isn't clear in the current wording.

Response: Thank you for your suggestion. We have reviewed the sentence and carefully paraphrased the sentence and cited the relevant source. The updates can be found on lines 417-419 of the revised manuscript.

Reviewer #1:

Dear Editor in Chief

I sincerely appreciate the opportunity to review the manuscript titled “Relationship of parenting styles on depression, anxiety, stress, and self-esteem of adolescents”, conducted in a socially and culturally diverse context such as Nepal.

Overall, I have observed that, in the introduction, method, data analysis, results, discussion, and conclusions, the authors have substantially improved the manuscript by carefully addressing...This study presents important and relevant research...Thank you for your thorough work, and congratulations to the authors for their dedication and the quality of this research.

Response: We are thankful to Reviewer #1 for their generous and encouraging comments. We are very happy that our previous revisions were found to be satisfactory and have strengthened the manuscript.

Reviewer #3:

Comment 1: Sample size, can the authors indicate, whether on average it was expected to recruit approx 58 participants from either public or private - would you expect a similar distribution or more in one type than another. Only because there is statement regarding exclusion of school that had <50 student, would this contribute to some selection bias?

Response: We thank the reviewer for this observation. We want to clarify the final sampling distribution resulting from proportionate sampling based on the total number of students in each randomly selected school, not an equal number from each school. This indicates that public schools in the municipality have larger enrollments. For clarification, we have mentioned this on lines 123-125 of the revised manuscript. Likewise, the exclusion of schools with <50 students was mainly due to logistical feasibility to ensure we reach the desired sample size within a manageable number of clusters. This might be subject to some bias, limiting the generalizability of findings to the adolescents within such settings. We have included this in the limitations of the revised manuscript as “Furthermore, the exclusion of schools with very few enrollments may limit…”

Comment 2: For ease of reading, suggest to include in the text, what p and q is? How was the design effect obatained and what was the variance cluster

Response: Thank you for the suggestion. We have now included the meaning of p and q in line 117 of the main manuscript. Similarly, the value for design effect (1.5) was chosen based on the commonly used estimate in many public health studies, as the actual ICC cannot be known until the data is collected.

Comment 3: The essence of univariate analyses, if the main goal is to understanfding which confounding variables go into the multivariate model, should be to assess significance. NOt sure assessing odds only in univariate is appropriate. Line 225, 226 - can this include line 229.

Response: Thank you for this valuable comment. We have reviewed the lines and made changes accordingly.

Comment 4: Was potential mediators also looked into? i.e impact of income as a confounder on the parenting styles and adolescent mental health, or even together with influecing the type school you went to, i.e private or public?; considered as some exploratory analyses.

Response: Thank you for this insightful comment. We did check for direct associations for potential mediators; however, we didn’t run any moderation analysis. As mentioned earlier, our objective was more focused on direct associations of factors. This opens the pathways for future research and has been mentioned within the discussion section on lines 472-473.

Comment 5: The income reported in dollars is, that monthly or annual? it would be good to clarify? Does it 72% did not have an income? or the range is outside this. It would be good to give context, is this low, medium or high earnings?

Response: Thank you for the feedback. The reported income in dollars represents the monthly income of the family of respondents. We have mentioned this in line 240 of the revised manuscript. The remaining 72% fall into other income categories: 25.7% earn <107.28 USD, 25% earn 214.55-321.83 USD, and 21.3% earn >321.83 USD, which is presented in Table 1. Likewise, we used the income category from similar research in adolescents,[1] but there was no established standard for high, medium and low earnings.

Comment 6: Table 1: Report Range of Age ( students, mothers, fathers) in the table?

Response: Thank you for the feedback. We have now incorporated the range for the respective age groups in Table 1 of the updated manuscript.

Comment 7: Table 1: Education level, does illiterate mean (no education?)

Response: Thank you for the feedback. Yes, the word “illiterate” used in the manuscript refers to no formal education.

Comment 8: Table 1: include for words for ECA in table or footnote so reader is not searching for what ECA means.

Response: Thank you for the suggestion. We have included the word in the footnote.

Comment 9: Table 2: Helps to present ranges, assuming these are normally distributed, suggested to also present median (IQR) too.

Response: Thank you for your suggestion. We have now presented the median and inter-quartile range in Table 2 of the revised manuscript.

We believe addressing these points has substantially strengthened our manuscript. Thank you.

Sincerely,

Anjali Bhatt

---

## [Editor Report · Decision Letter 2]

21 Oct 2025

Relationship of parenting styles on depression, anxiety, stress and self-esteem of adolescents

PONE-D-24-44012R2

Dear Dr. Bhatt,

We’re pleased to inform you that your manuscript has been judged scientifically suitable for publication and will be formally accepted for publication once it meets all outstanding technical requirements.

Kind regards,

Laura Kelly, PhD

Division Editor

PLOS One
---

## [Editor Report · Acceptance letter]

PONE-D-24-44012R2

PLOS ONE

Dear Dr. Bhatt,

I'm pleased to inform you that your manuscript has been deemed suitable for publication in PLOS ONE. Congratulations! Your manuscript is now being handed over to our production team.

Kind regards,

on behalf of

Dr. Laura Hannah Kelly

Staff Editor

PLOS ONE